# Improving Measurement Range of a Swellable Polymer-Clad Plastic Fiber Optic Humidity Sensor by Dye Addition [note 1]

**DOI:** 10.3390/s22166315

**Published:** 2022-08-22

**Authors:** Yuta Shimura, Yutaka Suzuki, Masayuki Morisawa

**Affiliations:** 1Graduate Faculty of Interdisciplinary Research, University of Yamanashi, Kofu 400-8510, Japan; 2Department of Biomedical Engineering, Faculty of Science and Technology, Toyo University, Tokyo 112-8606, Japan

**Keywords:** humidity sensor, POF sensor, swellable polymer cladding, dye-doped, optical fiber, leakage mode, Fresnel reflection

## Abstract

Humidity measurement is required in various fields. We previously developed a sensor that leverages the sudden change in the transmitted light intensity when switching from leakage mode to waveguide mode. By adjusting the low-refractive-index polymer of the cladding, we achieved measurements at 60% RH. However, for practical use, measurements at low humidity are essential. Therefore, in this study, we developed a sensor using a leakage mode that enables measurements at low humidity. To measure the leakage mode, it is necessary to increase the absorbance of the cladding and the incident angle at the core–cladding interface. Therefore, we developed a sensor in which the core was stretched, and the cladding was doped with a high concentration of dye. The experimental results confirmed that a sensor with a polymer concentration of 4% and a dye concentration of 3% could measure from 0% RH to 95% RH. The sensitivity was 0.1 dB/% RH from 0% RH to 70% RH and 0.32 dB/% RH from 70% RH to 95% RH. The estimated response time for a change from 10% to 90% light transmission for a sensor with 4% polymer concentration and 0.5% dye concentration was 22 s from 45% RH to 0% RH and 50 s from 0% RH to 45% RH.

## 1. Introduction

Currently, humidity measurement is required in various fields such as medicine, food, and precision equipment manufacturing and storage, and several measurement methods have been developed [1,2,3,4,5]. Commonly used electrical sensors that utilize capacitance or resistance are widely used because of their wide detection range, good reproducibility, and durability. Recently, carbon-based materials, such as carbon nanotubes and graphene oxide, have been developed as sensing materials and electric sensors manufactured by printing technology. For example, Chen et al. reported a humidity sensor using a composite thin film of reduced graphene oxide and lignosulfonic acid as sensing materials [6]. The sensor has a high response, low hysteresis, and stable reproducibility over a wide range of relative humidities from 22% to 97%. Shaukat et al. proposed a highly sensitive and fast ZrSe2-based humidity sensor via printing technology [7]. This sensor exhibited fast transient responses of 1 and 2 s, good stability over 360 min, and a linear relative humidity range of 15–80% RH. However, these electrical sensors face several challenges in harsh environments, such as susceptibility to electromagnetic noise and the need for a power supply near the sensing unit. Therefore, a humidity sensor using optical fibers was proposed. Fiber-optic sensors can be used in environments where electrical sensors cannot be used, because of their resistance to electromagnetism, corrosion, explosion, telemetry, and other features not found in electrical sensors [8,9,10].

Fiber optic humidity sensors can be classified into optical absorption, fiber Bragg grating, and interference types, according to their operating principles. For example, Zhao et al. proposed a Fabry–Perot-type fiber optic humidity sensor with a polyimide film coated on the tip of a multimode fiber as an interference-type sensor [11]. This sensor measures humidity by measuring changes in the reflected light intensity due to changes in film thickness. Fiber Bragg gratings are optical structures with periodic disturbances in the refractive index of waveguides. The sensor senses humidity by inducing a shift in the Bragg wavelength due to a change in the volume of the humidity-sensitive film. Several sensors based on this type of structure have been reported. For example, Li et al. proposed a highly sensitive FBG humidity sensor coated with polyimide and graphene films. This sensor exhibited high humidity sensitivity, with experimental results of 1.80 times that of a sensor coated with only a polyimide film [12]. Thus, fiber optic sensors that can adapt to environments where electrical sensors cannot be used have been widely studied [13,14,15].

We utilize plastic optical fibers with cores constructed from plastic. Plastic optical fiber (POF) is an optical fiber with a core and cladding constructed from polymer plastic that has the advantages of flexibility and lower cost compared to glass optical fibers. We developed a light-absorbing POF humidity sensor that uses polyvinylpyrrolidone (PVP), a polymer that swells with moisture, as the cladding for the POF [16]. PVP is a nonionic, water-soluble polymer that is used in a wide range of applications and fields such as adhesives and pigment dispersants due to its solubility in various organic solvents, high hygroscopicity, and film-forming and adhesive properties. In this sensor, PVP is used for cladding, taking advantage of its high hygroscopicity and film-forming properties. In this sensor, the operation of the POF is converted from the leakage mode to the waveguide mode as the refractive index decreases due to swelling. In the leakage mode region, the transmitted light intensity only slightly decreases with increasing humidity. However, when switched to the waveguide mode, the transmitted light intensity increases significantly with slight humidity changes. For this reason, humidity measurements have thus far used humidity levels at which the mode switches from leakage mode to waveguide mode as a starting point [17,18]. For the mode to easily switch from low humidity, the refractive index of the cladding should be close to that of the core. This can be realized, for example, by mixing a low-refractive-index polymer in the cladding, which has been used for measurements in humidity ranges above 60% RH. However, considering practical applications, it is essential to measure humidity below 60% RH [19].

To enable this POF sensor to measure in the low-humidity range, active use of the leakage mode can be considered. In the leakage mode, some light is reflected at the core-cladding boundary, owing to the refractive index difference and the angle of incidence. However, some of the light leaking into the cladding is reflected at the cladding–air interface and returns to the core, resulting in a small change in the transmitted light intensity in the leakage mode. Therefore, in this study, we propose a POF humidity sensor in which a cladding with a high concentration of dye is applied to a POF with the shape corrected by heating and stretching. As a result, the change in transmitted light intensity with the change in refractive index of the cladding in the leakage mode is increased. We have previously confirmed that dye doping allows for measurements in low humidity [20]. Extending that work, this study examines the effects of dye doping and sensor curvature, as well as changes in the POF sensor sensitivity with the length of the sensing part.

## 2. Principle of Operation of Swellable Polymer-Clad POF Humidity Sensor

### 2.1. Structure and Principles

Certain polymers exhibit swelling properties in the presence of certain substances. One such polymer, PVP, can swell selectively in water, reducing its refractive index from 1.52 to 1.49. This is because when water diffuses into PVP, the bonds between polymer molecules expand through the water, causing the volume of the polymer to expand. This volume expansion causes the molecular density of the polymer to become sparse and the polarizability to decrease. This is because the dielectric constant also decreases accordingly. Other polymers that swell in response to water include polyvinyl alcohol and hydroxyethylcellulose, among which PVP is characterized by its fast reaction rate and rapid decrease in refractive index at high humidity. Figure 1 shows the sensor structure and operating principle of the PVP and polyvinylidene fluoride mixed polymer (polyvinylidene difluoride: PVDF) cladding POF humidity sensor.

The cladding was doped with Brilliant Blue (BB), a dye used to increase absorbance. The core was constructed from polymethyl methacrylate (PMMA, refractive index n_1_ = 1.489), which is an acrylic resin that combines high transparency and impact resistance, and is a plastic used in a variety of products, including aquariums and airplane windows. Leveraging the fact that the intensity of the transmitted light changes significantly when the fiber behavior is converted from the leakage mode to the waveguide mode in the presence or absence of humidity, the ratio of the blend polymers was adjusted such that the refractive index n_2_ of the cladding polymer in the unswollen state was slightly higher than that of the core. The transmitted light intensity in the leakage mode is shown in Equation (1) using the ray-tracing method
(1)$$P_{out}=\overset{\pi /2}{\underset{\theta =0}{\sum }}P\left(\theta \right){\left\{R\left(\theta \right)+\left(1-R\left(\theta \right)\right)\cdot \alpha \left(c,\theta \right)\right\}}^{m\left(\theta \right)}$$
where *m*(*θ*) is the number of times a ray is reflected, and *α* is the transmittance of light incident on the cladding back to the core, which depends on the dye concentrations *c* and *θ*. Since this model uses the ray-tracing method, the reflectance *R*(*θ*) can be approximated by Equation (2) from the power reflection coefficient of Fresnel reflection used for plane waves.
(2)$$R\left(\theta \right)=\frac{{\left(n_{1}cos\theta_{i}-\sqrt{n_{2}{}^{2}-n_{1}{}^{2}sin^{2}\theta_{i}}\right)}^{2}}{{\left(n_{1}cos\theta_{i}+\sqrt{n_{2}{}^{2}-n_{1}{}^{2}sin^{2}\theta_{i}}\right)}^{2}}$$
where *θ_i_* is the angle of incidence at the boundary.

The second term in Equation (1) represents the effect of dye doping in the cladding. As the dye concentration *c* is increased, the amount of light absorbed by the cladding increases, resulting in a decrease in the transmittance *α* of the cladding. A decrease in *α* results in a decrease in the transmitted light intensity in the leakage mode.

The transmitted light intensity in the waveguide mode is expressed by Equation (3):(3)$$P_{out}=\overset{\theta_{cs}}{\underset{\theta =0}{\sum }}P\left(\theta \right)+\overset{\pi /2}{\underset{\theta =\theta_{cs}}{\sum }}P\left(\theta \right){\left\{R\left(\theta \right)+\left(1-R\left(\theta \right)\right)\cdot \alpha \left(c,\theta \right)\right\}}^{m\left(\theta \right)}$$
where *θ_cs_* denotes the critical angle. Comparing Equations (1) and (3), it can be seen that the transmitted light intensity increases rapidly when switching from the leakage mode to the waveguide mode. In the conventional method, the large change in the transmitted light intensity when switching from the leakage mode to the waveguide mode has been used for measurement. Therefore, by reducing the transmitted light intensity in the leakage mode, the ratio of the transmitted light intensity in the waveguide mode and leakage mode can be increased, thereby improving the sensitivity. Therefore, we improved the sensitivity by doping the cladding with the dye. However, when the absorbance at a certain wavelength is increased by adding the dye, the refractive index at that wavelength also increases. Therefore, the refractive index difference between the core and cladding increases at this wavelength, and the leakage mode humidity range becomes wider. Therefore, the mode-switching start humidity, which can be used for measurement, increases.

Therefore, in this study, we propose the use of leakage mode for measurements at low humidity. Thus, the first term in Equation (1), i.e., the change in the reflectance in Equation (2) was used for measurement. To realize the measurement in the leakage mode, *θ_i_* in Equation (2) should be increased, and the second term in Equation (1) should be decreased. Therefore, the curvature of the POF was removed, and the dye was doped. The reasons for this are as follows.

### 2.2. Effect of POF Curvature

Since the POF is wound and housed on a shaft of diameter 20 cm, it is slightly curved in its natural state with a radius of curvature of approximately 5.5 m^−1^. In previous sensors, this curvature could be ignored, because the measurement was made in the waveguide mode, and the effect of the refractive index difference between the core and cladding was larger than the incident angle *θ_i_* on the core-cladding boundary. However, since the present measurement was performed in the leakage mode, the angle of incidence to the core-cladding boundary directly affected the magnitude of the transmitted light intensity, as shown in Equation (2). Therefore, we used a POF with a straight core shape by stretching the fiber and heating it. By making the core shape closer to the ideal straight core shape, the transmitted light intensity in the leakage mode increases, and the rate of change of the transmitted light intensity with respect to the change in cladding refractive index also increases.

It is difficult to keep the curvature of the sensor constant. By adding curvature to the sensor, the closer the refractive index of the cladding is to the core, the more light leaks into the cladding, resulting in a relatively high sensitivity based on the lowest transmitted light level. However, if the curvature cannot be kept constant during measurement, the sensitivity of the sensor will change. In addition, the curvature compromised the uniformity of the cladding during the sensor creation. As a result, the reproducibility of humidity characteristics was reduced due to the blurring of the curvature. However, by eliminating the curvature and making the sensor straight, it is possible to improve the uniformity of the cladding, suppress variations in sensitivity during measurement, and improve sensor reproducibility.

### 2.3. Dye Doping

As described previously, doping with a dye increases the optical absorption for a given wavelength, and at the same time, increases the refractive index. Doping the dye such that the refractive index of the cladding is significantly higher than that of the core expands the range of leakage modes and extends the range of possible measurements. In addition, since sensors using the leakage mode measure Fresnel reflection, the greater the difference in refractive index between the core and cladding, the greater the change in transmitted light intensity. In other words, the measurement range and sensitivity can be increased by doping with large amounts of dye.

## 3. Construction of the Leaky Waveguide POF Humidity Sensor

The POF sensor was fabricated as follows. First, POF (Eska SH-4001) (PMMA: refractive index 1.489) with a jacket core diameter of 0.98 mm was cut into 15-cm-long pieces. Next, to stretch the POF in a straight line, a shaft and crank arm combination device was used to fix both ends of the POF and stretch it. With the POF fixed in a straight line, the POF was heated at 80 °C for 2 min, cooled to room temperature, and then the curvature and distortion of the POF was removed. Since marks were left at both ends after these processes, the ends were cut at 1 cm intervals to remove the marks. Next, the jacket was removed from the center of the 9 cm section, soaked in dioxane for 3 min, and then wiped off with an industrial paper wiper (Kim wipe) to remove the cladding originally attached to the POF. Next, a PVP:PVDF mixed dimethyl sulfoxide solution containing BB dye (mixing ratio 5:2) was dip-coated onto the stretched PMMA core. The pull-up speed was set at 5 mm/s. The cladding was dried at room temperature in air for 24 h. PVDF is a low-refractive-index polymer (refractive index 1.42) and is used to adjust the refractive index of the cladding and form a uniform cladding on the core. Dimethyl sulfoxide, a common solvent for PVP and PVDF, was used as the solvent. PVP and PVDF were mixed in dimethyl sulfoxide on a mass basis. The solution was stirred in a stirrer for 24 h to achieve a homogeneous solution.

## 4. Measurement Results and Discussion on Leakage Waveguide Type POF Humidity Sensor

The sensor measurements were performed using the experimental system shown in Figure 2. A laser beam (DPS-2002, NEOARK Corp., Tokyo, Japan) with a wavelength of 650 nm was connected to a photosensor amplifier (C6386-01, HAMAMATU PHOTONICS K.K., Hamamatu, Japan) using a sensor head. A digital multimeter (NR500, KEYENCE, Tokyo, Japan) was connected to read the voltage output from the optical sensor amplifier, and humidity characteristics were reviewed on a PC. A reference humidity sensor (MR6662, CHINO, Tokyo, Japan) was placed inside the chamber. A bottle filled with water was prepared outside the chamber and filled with saturated water vapor by pumping air into the water using a pump. The bottle was connected to the chamber with a tube to feed moist air into the chamber and was controlled by adjusting a valve.

The light from the laser source was connected directly to a fiber and propagated in multiple modes. The sensor was fixed to the shaft to prevent it from bending during measurement. The chamber was 16 cm wide, 16 cm deep, and 8 cm high. At the start of the measurement, nitrogen gas was pumped into the chamber to reduce the humidity in the chamber to 0% RH. Nitrogen does not affect the sensor because it accounts for approximately 78% of air. After confirming that the humidity in the chamber had reached 0% RH, measurements were started with the reference humidity sensor and the optical sensor amplifier, and the humidity characteristics were measured with the valve adjusted so that the humidity in the chamber increased at 1% RH per min, thus making the humidity in the chamber uniform.

Figure 3 shows the change in the transmitted light intensity versus humidity for the sensor measured without thermal stretching (bending) and for the sensor measured after removing curvature and distortion by thermal stretching (straightening). The vertical axis is the output of the photosensor amplifier, which is the voltage value of the transmitted light intensity value obtained from the photodiode and amplified by the amplifier. The humidity characteristics of the bend-type sensor show that the change in transmitted light intensity in the leakage mode is small. This indicates that the bend-type sensor is suitable for measurements at high humidity. Moreover, the linear-type sensor has a large angle of incidence of light entering the core–cladding interface, resulting in less light absorption by the cladding and an overall increase in the transmitted light intensity. This reduces the likelihood that all light will leak out during leakage mode, indicating that leakage mode can be used. In addition, since the fiber geometry is closer to an ideal shape, changes in the leakage modes can be clearly observed.

Adding dye to the cladding improves the absorbance of the cladding and increases its refractive index relative to the wavelength of the laser light. Figure 4 shows the humidity characteristics of a sensor with a cladding composition of 6.8% polymer concentration and 0.2% BB concentration. The transmitted light intensity on the vertical axis is a graph in which the overall light intensity is converted to a sensitivity with the lowest transmitted light intensity as 1.

The addition of 0.2% BB dye to the cladding significantly improved the change in the transmitted light intensity in the leakage mode. The conversion from leakage mode to waveguide mode approached approximately 70% RH from 60% RH, and the leakage mode region was significantly expanded. The sensitivity was 0.22 dB/% RH between 20% RH and 70% RH, which was higher than that of a tungsten dioxide-coated optical absorption fiber sensor (0.1213 dB/% RH between 35% RH and 85% RH) [21] and a U-shaped bent optical fiber coated with silica film and doped with methylene blue (0.087 dB/% RH between 1.1% RH and 4.1% RH) [22]. This is because the addition of BB dye to the cladding increased the refractive index of the cladding for light at a wavelength of 650 nm of the laser light source used in the measurement, and also increased the absorbance of the cladding. The higher absorbance of the cladding reduced the amount of light that leaked into the cladding and returned to the core, rendering the changes in light reflected at the core–cladding interface clearly visible. As the refractive index difference between the core and cladding increased, the reflectance change at the core–cladding interface increased, further expanding the leakage-mode region.

Figure 5 shows the humidity characteristics of a sensor comprising a cladding with a polymer concentration of 4% and BB dye concentration of 3%, measured at 0% RH.

As the concentration of the BB dye increased, the refractive index of the cladding with respect to the 650 nm wavelength became larger than that of the core, resulting in a leakage mode between 0% RH and 95% RH. The sensitivity was as high as 0.1 dB/% RH from 0% RH to 70% RH and 0.32 dB/% RH from 70% RH to 95% RH. The high sensitivity from 70% RH and up can be attributed to the PVP property that its refractive index changes rapidly under high humidity. These results indicate that the sensor can measure from 0% RH, confirming that a sensor capable of measuring from 0% RH to 95% RH was fabricated.

The relationship between the length of the sensing section and humidity characteristics is shown in Figure 6. Sensors with sensing part lengths of 3, 6, and 9 cm were measured for coatings comprising a polymer concentration of 3.9% and BB concentration of 0.1%. The sensitivity of each sensor from 0% RH to 60% RH was 0.12 dB/% RH, 0.09 dB/% RH, and 0.05 dB/% RH, with the sensitivity increasing with the length of the sensing section. This is because the longer the sensing section, the higher the number of reflections and the greater the change in transmitted light intensity.

Figure 7 shows the time response of the sensor with a polymer concentration of 4% and BB concentration of 0.5%, showing the change in transmitted light intensity when the humidity was lowered from 45% RH to 0% RH and when it was raised from 0% RH to 45% RH. The humidity change was determined by closing the chamber lid, rapidly introducing nitrogen gas to lower the humidity from 45% RH to 0% RH, and opening the lid to raise the humidity to 45% RH. When the humidity was lowered from 45% RH to 0% RH, the time required for the transmitted light intensity to stabilize was approximately 80 s. When the humidity was raised from 0% RH to 45% RH, the time until the transmitted light intensity stabilized was approximately 150 s. The approximate response times from 10% to 90% transmitted light intensity were 22 s from 45% RH to 0% RH and 50 s from 0% RH to 45% RH.

## 5. Conclusions

To measure humidity over a wide range, including low humidity, we developed a sensor using the leakage mode. The core shape was straightened to increase the angle of incidence at the core–cladding interface, and the dye was doped into the cladding to increase the refractive index difference between the core and cladding and the absorbance of the cladding. This increased the change in the transmitted light intensity in the leakage mode, which enabled measurements from low-humidity conditions. The experimental results confirmed the change in the light intensity in the leakage mode by linearizing the core geometry. In addition, the addition of the dye broadened the leakage mode region and improved the sensitivity. Furthermore, it was confirmed that the sensor with a polymer concentration of 4% and a BB concentration of 3% could measure in the humidity range from 0% RH to 95% RH, with a sensitivity of 0.1 dB/% RH between 0% RH and 70% RH and 0.32 dB/% RH between 70% RH and 95% RH. These results indicate that a sensor with a measurement range of 60% RH or higher can be improved to a sensor capable of measuring from 0% RH to 95% RH. Next, the variation in the transmitted light intensity with the length of the detector element was checked, and it was confirmed that the longer the detector element, the greater the variation in the transmitted light intensity. Finally, the response time of the sensor was checked. The sensor had an estimated response time of 22 s for a decrease in humidity from 45% RH to 0% RH and 50 s for an increase in humidity from 0% RH to 45% RH. In future work, we will consider improving the sensitivity in the leakage mode by increasing the amount of light absorbed by the cladding by applying precise microbending to increase the change in transmitted light intensity in the leakage mode while maintaining reproducibility. We also plan to compare PVP with other swellable polymers to see if PVP provides an advantage in reaction speed in sensors that measure in leakage mode.

## Figures and Tables

**Figure 1 sensors-22-06315-f001:**
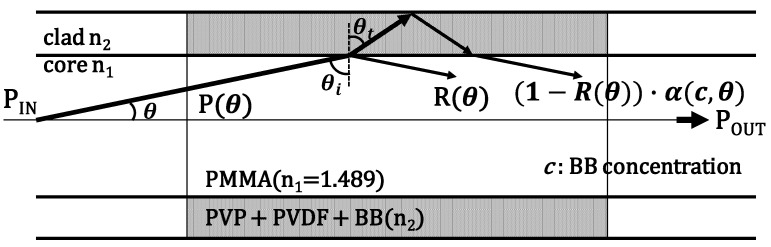
POF sensor model.

**Figure 2 sensors-22-06315-f002:**
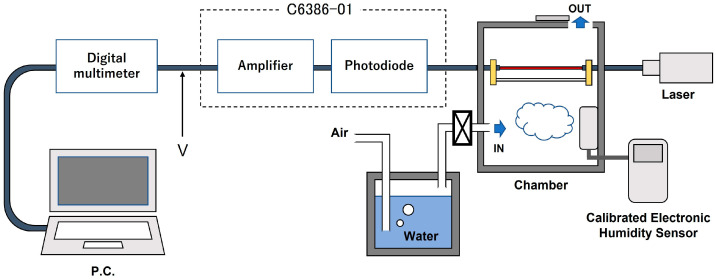
Experimental system for measurements with the POF sensor.

**Figure 3 sensors-22-06315-f003:**
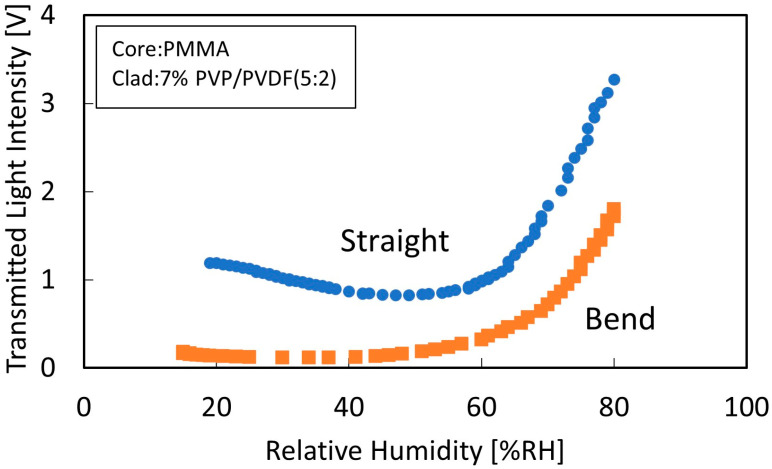
Humidity characteristics of sensors without stretching treatment and linear sensors.

**Figure 4 sensors-22-06315-f004:**
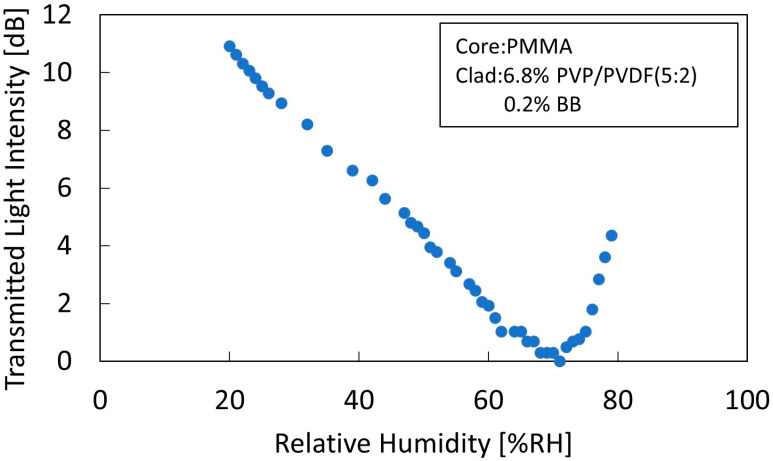
Humidity characteristics of a leaky waveguide sensor comprising a cladding with a polymer concentration of 6.8% and a BB concentration of 0.2%.

**Figure 5 sensors-22-06315-f005:**
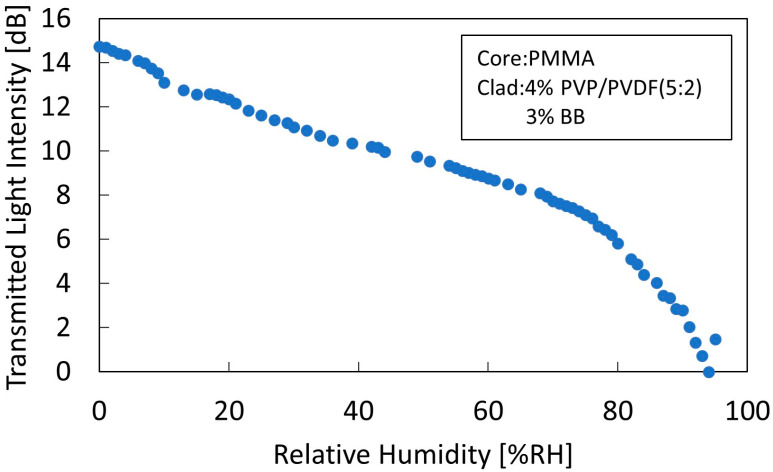
Humidity characteristics of a sensor comprising a cladding with a polymer concentration of 4% and a BB concentration of 3%, measured at 0% RH humidity.

**Figure 6 sensors-22-06315-f006:**
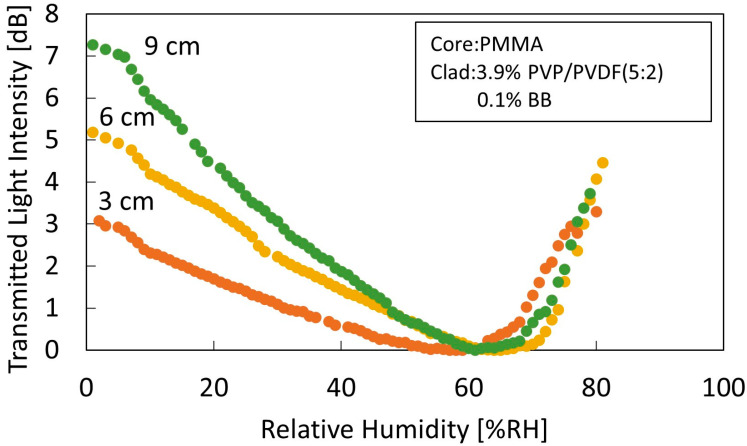
Comparison of humidity characteristics of sensors of length 3 cm, 6 cm, and 9 cm of the sensing area comprising a cladding with a polymer concentration of 3.9% and BB 0.1%.

**Figure 7 sensors-22-06315-f007:**
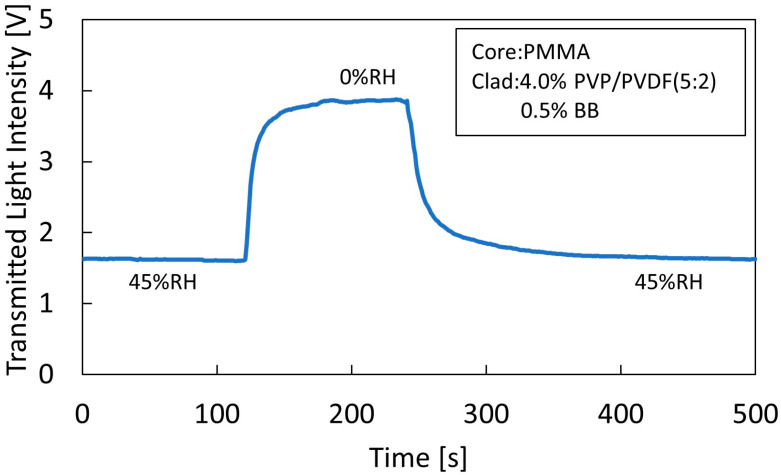
Response time from 45% RH to 0% RH and 0% RH to 45% RH for sensors with 4% polymer concentration and 0.5% BB concentration.

## Data Availability

Not applicable.

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
