# Peer review of "Improving Measurement Range of a Swellable Polymer-Clad Plastic Fiber Optic Humidity Sensor by Dye Additionâ€"

_sensors, 2022, doi:10.3390/s22166315_

Round 1
Reviewer 1 Report
Dear editor,
In this work, authors have post-processed a polymer optical fiber in order to measure humidity. The do so, they have prepared a new polymer cladding material. The principle behind measurement is based on light leakage due to refractive index change of the polymer material associated with the swelling effect.
Overall, I consider that the work is innovative and well written. However, In my opinion, the manuscript needs some improvements prior to publication.
· Authors need to describe the work in terms of sensitivity, limit of detection and compare their performance with the results already published in the literature.
· It is necessary to justify the reason why the refractive index of the doped cladding decreases when water diffuses into the polymer.
· Works developed on thinner polymer fibers (~100um), describe tens of minutes for water to completely diffuse into the polymer matrix and stabilize the signal. In this work authors use a fiber (ESKA) with a diameter 10 times higher. What is the necessary time to stabilize the signal? Is there any graph to show the stabilization time?
· It would be interesting to comment on the possibilities to use other types of fiber optic sensors based on polymer materials, such as: "Simultaneous detection of humidity and temperature through an adhesive based Fabry – Pérot cavity combined with polymer fiber Bragg grating”.
· The results presented in this work are described in a.u. It is difficult to quantify the power changes without a proper scale. I would recommend the authors to show the results in proper units such as mW, dBm, dB, %...
Based on the above, my recommendation is to revise the manuscript according to the above descriptions.
Author Response
We thank the reviewer for their suggestions and comments for our manuscript. We have revised our manuscript and added some explanations, mainly in accordance with the suggestions. We look forward to the favorable consideration of the reviewer.
Revisions are shown in the attached file.

Reviewer 2 Report
In this manuscript, the authors developed a humidity sensor that leverages the sudden change in the transmitted light intensity when switching from the leakage mode to the waveguide mode using an optical fiber. Below are some questions and comments that need to be addressed.
1. Bending the optical fibers in different directions, different angles sometimes lead to changes in light intensity, polarity, etc. How this affect the sensing results?
2. When setting up the experiments in Figure 2, there seems to be a distance in the calibrated electronic humidity sensor and the optical sensor. Is the humidity increase in the chamber homogeneous?
3. What is underlying physical mechanism showing bended optical fiber is more suitable than straight fiber or this is just an empirical observation?
4. How would the clad composition change the sensing sensitivity?
5. The authors show by increasing the sensing length, the sensitivity increases. Will the sensitivity continue to increase with sensing length or this trend will eventually plateau?
Author Response

(The authors gave the same response as above.)

Reviewer 3 Report
This study investigates a swellable polymer clad palstic fiber to monitor humidty through dye addition. It is well constructed. The following questions need to be answered before publish.
1. The core was stretched during measurement, but how to install the humidity in reality if there is no equipment to stretch it.
2. "in future work, we willl consider improving the sensitivity in the leakage mode", how do you improve the sensitivity?
3. What is the stability of the sensor in reality?Will the dye leach out in the long run?
Author Response

(The authors gave the same response as above.)

Round 2
Reviewer 1 Report
Dear editor,
The authors have revised the manuscript acordingly. Thus, I woul like to congratulate the authors and recommend the publication of the manuscript.
Best regards
Reviewer 3 Report
Accepted.